# Therapeutic Potential of Photobiomodulation for Chronic Kidney Disease

**DOI:** 10.3390/ijms23148043

**Published:** 2022-07-21

**Authors:** Ji Bian, Ann Liebert, Brian Bicknell, Xin-Ming Chen, Chunling Huang, Carol A. Pollock

**Affiliations:** 1Kolling Institute, Sydney Medical School, Faculty of Medicine and Health, University of Sydney, Royal North Shore Hospital, St Leonards, NSW 2065, Australia; jbia3972@uni.sydney.edu.au (J.B.); xin-ming.chen@sydney.edu.au (X.-M.C.); 2Faculty of Medicine and Health, University of Sydney, Camperdown, NSW 2006, Australia; ann@pbmconsults.com; 3College of Health and Medicine, Australian National University, Deacon, ACT 2600, Australia; brian@pbmconsults.com

**Keywords:** photobiomodulation, chronic kidney disease, mitochondrial dysfunction, oxidative stress, inflammation, gut microbiota dysbiosis

## Abstract

Chronic kidney disease (CKD) is a growing global public health problem. The implementation of evidence-based clinical practices only defers the development of kidney failure. Death, transplantation, or dialysis are the consequences of kidney failure, resulting in a significant burden on the health system. Hence, innovative therapeutic strategies are urgently needed due to the limitations of current interventions. Photobiomodulation (PBM), a form of non-thermal light therapy, effectively mitigates mitochondrial dysfunction, reactive oxidative stress, inflammation, and gut microbiota dysbiosis, all of which are inherent in CKD. Preliminary studies suggest the benefits of PBM in multiple diseases, including CKD. Hence, this review will provide a concise summary of the underlying action mechanisms of PBM and its potential therapeutic effects on CKD. Based on the findings, PBM may represent a novel, non-invasive and non-pharmacological therapy for CKD, although more studies are necessary before PBM can be widely recommended.

## 1. Introduction

Chronic kidney disease (CKD), a condition characterized as a persistent kidney damage or glomerular filtration rate (GFR) <60 mL/min/1.73 m^2^ for more than three months, has increased dramatically in recent years, with a global prevalence of between 8 and 16% [1,2]. As CKD is often asymptomatic in its early stages, less than 5% of patients with early stage CKD are aware of their disease [3]. There are 5 stages of CKD, based on glomerular filtration rate and albuminuria, irrespective of the cause of CKD [3]. In addition to the high risk of progression to end-stage kidney disease (ESKD), CKD also increases cardiovascular complications, the leading cause of morbidity and mortality in CKD [4]. For decades, much research has investigated well-accepted pathogenic factors contributing to CKD onset and progression, such as obesity, hypertension, dyslipidemia, and diabetes mellitus. Accumulated evidence has suggested that pathogenic mechanisms, including mitochondria dysfunction, oxidative stress, inflammation, and dysregulation of the gut microbiome, play pivotal roles in the development and progression of CKD. The implementation of current evidence-based clinical practices succeeds in only delaying the development of kidney failure. Death, transplantation, or dialysis are the consequence of kidney failure, resulting in a significant burden on the health system. There is an urgent demand for novel therapies for CKD treatment.

PBM, previously known as low-level laser (light) therapy (LLLT), uses visible and near-infrared light with a wavelength from 450 to 1100 nm to trigger photochemical changes within intracellular cellular structures. PBM has been used clinically for more than 50 years and studied in various diseases and conditions, such as muscle [5], wound healing [6], pain relief [7], arthritis [8], hair loss [9], orthodontic mini-implant stability [10], neurodegenerative diseases [11] and most recently in COVID-19 patients [12]. PBM has been shown over decades to be a patient-directed, adjunctive, effective, non-invasive, and safe treatment option to mitigate mitochondrial dysfunction [13], oxidative stress [14,15], inflammatory conditions [16], and, more recently, gut microbiota dysbiosis [17], which are all primary contributors to CKD. Notably, several studies have demonstrated a positive effect of PBM in different models of kidney diseases, including acute kidney injury, glomerulonephritis, and metabolic syndrome-related kidney injury [18].

This review focuses on the major pathogenic factors contributing to CKD and the beneficial effects of PBM by targeting those mechanisms.

## 2. Primary Pathogenic Mechanisms of CKD

### 2.1. Mitochodrial Dysfunction and Oxidative Stress in CKD

Mitochondria serve as the cellular “powerhouses” of cells, synthesizing adenosine triphosphate (ATP). They are also a potent mediator of primary intracellular reactive oxygen species (ROS). Maintaining mitochondrial function and structure is critical to cellular metabolism, homeostasis, and survival [19]. The kidney, a high-energy demand organ requiring large quantities of ATP energy to actively maintain its normal function [20], is rich in mitochondria. Hence, the kidney is susceptible to mitochondrial dysfunction, which is increasingly recognized to play a pivotal role in the progression of CKD [21]. Mitochondrial defects, including a decrease in the quality and quantity of mtDNA, have been documented in experimental models and in patients with CKD [22,23,24,25]. In addition, the accumulation of damaged mtDNA and fragmented mitochondria leads to mitochondrial dysfunction, which is a hallmark of tubular injury, particularly in patients with diabetic kidney disease (DKD) [26]. Similarly, lower mitochondrial content and reduced mitochondrial proteins, including the mitochondrial transcription factor A (TFAM), an essential regulator of the mitochondrial genome, have been reported in patients with CKD and in mouse models of folic acid (FA) and unilateral ureteric obstruction (UUO)-induced CKD [27]. Overexpression of carnitine palmitoyl-transferase 1A (CPT1A), the fatty acid shuttling enzyme involved in the fatty acid oxidation, alleviated kidney fibrosis and improved kidney function in three experimental models of CKD by enhancing fatty acid oxidation and restoring impaired mitochondrial function with increased mitochondrial mass, and normalized bioenergetics and ATP production [28].

In addition, mitochondrial dysfunction also contributes to CKD progression via oxidative stress which occurs due to an imbalance between the overproduction of mitochondrial reactive oxygen species (mtROS) and reduced antioxidant systems [29]. Increased oxidative stress has been observed in CKD and dialysis patients [30]. In the kidney, excessive mtROS production, mainly produced by deregulated mitochondrial respiratory chain via oxidative phosphorylation system, cannot be removed by antioxidant systems and consequently leads to oxidation of cellular components such as DNA, proteins, and lipids, resulting in kidney cells damage [31,32]. For example, Huang et al. reported oxidative stress damage in DKD rats, evidenced by increased expression of malondialdehyde and decreased superoxide dismutase (SOD). In addition, researchers indicated that treatment with Dencichine, an active component in Chinese medicinal herbs, attenuated diabetes-induced kidney injury in DKD rats by inhibiting oxidative stress responses, enhancing the antioxidant capacity, and normalizing autophagy as well as reducing renal cell apoptosis [33]. A clinical study identified that oxidative stress marker, including increased advanced oxidation protein product, myeloperoxidase, malondialdehyde, nitric oxide, oxidized low-density lipoprotein, and decreased glutathione, are closely related to the carotid atherosclerosis process in patients with CKD, indicating those oxidative markers may serve as the markers to predict the progression of CKD [34].

Therefore, mitochondrial dysfunction and oxidative stress derived from damaged mitochondria contribute to the pathogenesis of CKD. A therapeutic strategy targeting mitochondria to improve mitochondrial function or relieve oxidative stress is thus likely to be beneficial in improving kidney function in CKD.

### 2.2. Inflammation and CKD

Inflammation is characterized by the activation of a variety of inflammatory markers, such as cytokines, chemokines, and cell adhesion molecules, mainly produced by the innate immune system. Inflammation is triggered by injury during a complex biological preventive and reparative process. Many studies have elucidated that inflammatory molecules and signaling pathways are directly involved in the development and progression of CKD [24,35].

Cytokines are proteins of molecular weight between 15 and 20 kDa that interplay with the development and activity of the immune system, such as interleukins (ILs), tumor necrosis factor alpha (TNF-α), interferon (IFN-γ), and transforming growth factor-beta 1 (TGF-β1). They play an essential role in paracrine, autocrine, and endocrine signaling [36]. Interleukin 6 (IL-6) is one of the interleukins secreted by leukocytes and the most studied in kidney disease due to its pro-inflammatory effects. The serum level of IL-6 is significantly increased in CKD patients compared to healthy subjects [37]. Of interest, Tocilizumab, an IL-6 receptor-targeted drug, significantly reduced the glomerular and tubulointerstitial fibrosis via inhibiting the IL6/ERK signaling pathway in the UUO model of CKD [38]. Milas et al. found that increased levels of pro-inflammatory interleukins, such as IL-1α, IL-8, and IL-18, were associated with podocyte injury and proximal tubular dysfunction in the early stage of DKD in patients with type 2 diabetes mellitus [39]. Additionally, elevated serum IL-9 levels have been found in patients with primary focal and segmental glomerulosclerosis, suggesting that the up-regulated expression of IL-9 was induced by glomerular injury in humans [40]. TNF-α is secreted by invading immunologic cells, specifically by monocytes/macrophages [41]. A clinical study with 133 CKD patients found that TNF-α and INF-γ, Th1 (T helper cell type 1) cytokines significantly increased in CKD patients, suggesting that Th1 cells were activated in the inflammatory response induced by CKD [42]. It is well known that TGF-β1 is a key mediator of kidney fibrosis, with the TGF-β1 pathway shown to be modified by photobiomodulation [18]. Conversely, recent studies have identified TGF-β1 as a potent anti-inflammatory cytokine, which negatively regulates renal inflammation [43,44]. The anti-inflammatory effect of TGF-β1-partially explains the negative trials of TGF-β1 antibodies in human CKD [45]. The diverse roles of TGF-β1 in kidney fibrosis and inflammation has been well summarized [46].

### 2.3. Gut Microbiota Dysbiosis in CKD

Dysregulated composition and function of the gut microbiome, termed gut microbiota dysbiosis, has been recognized as a pathogenic mechanism in many diseases, including CKD [47]. Previous studies have shown that gut bacteria dysbiosis contributes to CKD via several mechanisms, such as accumulation of uremic toxins, decreased production of short-chain fatty acids (SCFAs), disturbed enteroendocrine, and leaky gut barrier [2].

Significant alterations in the gut microbiome composition, richness, diversity, and blood and fecal metabolic composition have been found in patients with CKD and kidney failure, supporting a crucial role of gut dysbiosis in the pathogenesis of CKD [48,49,50]. Transferring gut microbiota from patients with kidney failure into CKD models of germ-free mice and antibiotic-treated rats resulted in increased uremic toxins, enhanced oxidative stress, and aggravated renal fibrosis, confirming a causative contribution of the aberrant gut microbiota to CKD [51]. In a clinical study with 73 pre-dialysis and dialysis patients with kidney failure and 19 healthy controls, gut microflora was measured via high-throughput sequencing. The results indicated that hemodialysis, but not peritoneal dialysis, mitigated gut microbiota disorders by increasing the richness of beneficial bacteria and reducing some potential pathogenic bacteria compared to pre-dialysis patients with kidney failure [52].

Uremic toxins, produced by gut bacterial metabolism, typically accumulate at the early stage of CKD, and contribute to progressive kidney function loss [1]. Dysregulated gut bacteria in CKD are associated with increased production of gut-derived uremic toxins, including p-cresyl sulfate, p-cresyl glucuronide, indoxyl sulfate, indole acetic acid, and trimethylamine n-oxide (TMAO) in the gut lumen and blood circulation, which is facilitated by increased intestinal barrier permeability and impaired excretory function [53,54]. The study by Gryp et al. found increased protein-bound uremic toxins, including p-cresyl sulfate, p-cresyl glucuronide, indoxyl sulfate, and indole acetic acid, in plasma but not in feces and urine from CKD patients, indicating that the accumulated plasma protein-bound uremic toxins are likely due to the impaired kidney function [55]. In addition, increased circulating TMAO and its three main precursors, choline, betaine, and L-carnitine, were found to be closely correlated with the grade of CKD and estimated glomerular filtration rate, suggesting that the gut microbiota metabolite TMAO and its precursors together could be potential non-invasive biomarkers for CKD [56,57]. Similarly, Sun and his team found that elevated serum 3-indole propionic acid level, a microbial tryptophan metabolite, is negatively correlated with CKD development, indicating 3-indole propionic acid may be a key biomarker and protective factor for CKD [58].

SCFAs are derived from anaerobic bacterial fermentation of indigestible foods in the large intestine and then absorbed into the systemic circulation. The primary SCFAs produced in humans include acetate, propionate, and butyrate [59,60]. SCFAs regulate regional inflammation and cellular metabolism in kidneys via binding to G-protein coupled receptors (GPCRs) such as GPR43, GPR41, and GPR 109a [61], which are expressed on almost all immune cells [62]. Reduced levels of SCFAs were observed in patients with CKD [63,64]. For example, significantly lower levels of fecal SCFAs, including acetic acid, propionic acid, butyric acid, iso-butyric acid, and caproic acid, have been found in patients with IgA nephropathy, which is accompanied by altered gut microbiota diversity [63]. The beneficial effect of SCFAs has been further confirmed in different mouse models of CKD, including diabetic- and folic acid-induced CKD models [65,66]. A recent study highlighted the reno-protective effect of dietary fiber by promoting the expansion of the SCFAs-producing bacterial population, leading to increased SCFAs concentration. SCFA supplementation treatment further confirmed the direct beneficial effect of SCFAs in the CKD mice through SCFA mediated histone deacetylase activity and the GPR pathway [67].

Collectively, these data suggest that modifying the gut microbiota dysbiosis observed in patients with CKD may provide a new therapeutic strategy for treatment.

## 3. PBM Overview

PBM uses non-thermal light for therapeutic purposes via the modulation of cellular activities and responses. The therapeutic property of PBM was first discovered in the 1960s by Dr. Endre Mester [68], with clinical applications significantly increasing in recent years due to its broad range of biological activities and decades-long safety records [69,70,71]. Generally, PBM is performed with a low-powered laser or light-emitting diodes (LED). to trigger photochemical changes within cellular structures. The parameters of PBM that influence PBM’s effectiveness on the target cells and tissues include the light source, wavelength, power output, energy density, duration of irradiation, number of treatments, and the mode of light delivery (continuous or pulsed). For example, different wavelengths of light have different degrees of absorption, scattering coefficient, and reflection in the human body [72,73]. Near-infrared wavelengths have lower scattering factors and much higher penetration into tissues [74,75]. In addition, skin contact with the PBM source also increases penetration into tissues since reflection is reduced, resulting in greater power density in the target tissue [76]. Moreover, PBM in a pulsed mode has been suggested to result in improved therapeutic outcomes compared to continuous mode [77].

A critical feature of PBM is its biphasic dose response, described as “Arndt-Schulz Law”, where increasing levels of PBM stimuli accelerate the biological reactions until a peak is reached. After this peak, higher levels of PBM will have reduced or even a detrimental effect [78]. PBM dose is therefore critical for treatment outcomes. It is increasingly recognized that the biphasic dose response is responsible for a significant number of inefficacious or poor responses in previous PBM studies [79]. Clearly, optimal conditions for effective PBM doses in different conditions need to be defined and highlights the importance of specific parameters, which need to be tailored for preclinical in vitro, in vivo studies and then clinical use.

Although the action mechanisms underlying the beneficial effect of PBM are not fully understood, the most widely accepted mechanism is that PBM improves mitochondrial function, leading to increased ATP production, inhibiting oxidative stress, and reducing inflammation [15,80,81]. Furthermore, recent studies have demonstrated that PBM can modulate the gut microbiota and may be a novel gut microbiota intervention in different animal disease models, including Alzheimer’s disease and osteoporosis [17,82,83] and, more recently, in patients with Parkinson’s disease [84,85].

### 3.1. PBM and Mitochodrial Activity

PBM is widely considered to exert beneficial effects via improving mitochondrial function [86]. The accepted concept is that PBM works on the principle that light-sensitive molecules in the body known as chromophores are excited by photonic stimulation. It is generally recognized that the principal chromophore responsible for light absorption is complex IV (cytochrome c oxidase, CCO) of the mitochondrial electron transport chain [13]. The photon absorption by CCO can restore the rate of respiration, increase ATP production and improve cellular metabolism, which is likely to work through the photodissociation of the inhibitory nitric oxide from the enzyme [87]. Choi et al. applied PBM at 950 nm to mitigate ROS in the post-thawing injury of human stem cells from the apical papilla through stimulating CCO activity, leading to increased mitochondrial membrane potential and consequent higher amount of ATP production [88]. Furthermore, PBM (904 nm, 6.23 J/cm^2^) improved mitochondrial dynamics via modulating the expression of mitofusin 2 and dynamin-related protein 1 in type 1 diabetic rats [89], partially preventing or reversing mitochondrial dynamics dysfunction. PBM also preserved mitochondrial function and integrity in rats with neonatal hypoxic-ischemia injury, leading to significant alleviation or prevention of hypoxic-ischemia-induced brain injury [90].

### 3.2. PBM and Oxidative Stress

PBM can modulate the level of oxidative stress via regulating ROS and the antioxidant system based on the stress condition of the cells. PBM produces low levels of ROS in normal cells. However, when cells are exposed to excessive or prolonged oxidative stress, PBM increases mitochondrial membrane potential and reduces stress-induced ROS production [86]. In vitro, the protective effect of PBM against hyperglycemia-induced oxidative stress has been reported in adipose-derived mesenchymal stem cells extracted from streptozotocin-induced diabetic rats. PBM irradiation increased the expression of antioxidants, including SOD1, catalase (CAT), and total antioxidant capacity, but reduced ROS and oxidative stress biomarkers such as nicotinamide adenine dinucleotide phosphate oxidases 1 and 4 (Nox1 and Nox4) [91]. Similarly, in diabetic wounded fibroblast cells, PBM (660 nm, 5 J/cm^2^; or 830 nm, 5 J/cm^2^) significantly increased the expression of SOD, heme oxygenase, CAT and reduced the level of forkhead box protein 1, indicating that PBM improved diabetic wound healing by mitigating oxidative stress [81]. The beneficial effect of PBM in the acceleration of wound healing was further confirmed in a diabetic mouse model by improving the mitochondrial redox system [90]. In addition, recent studies showed that PBM prevented high-intensity resistance exercise-induced oxidative stress in female Wistar rats, and the application of PBM after exercise provided more evident benefits [92,93].

### 3.3. The Role of PBM on Inflammation

One of the well-known effects of PBM is an overall inhibition of inflammation, as evidenced in various inflammatory conditions, such as diabetic wounds [94], retinal diseases [95], asthma [96], and the aging brain [72]. PBM exerts its anti-inflammatory effect by reducing many cytokines and inflammatory mediators in different inflammatory conditions such as carrageenan-induced paw edema, blepharitis, and osteoarthritis [10,97,98]. In human periodontal ligament fibroblasts stimulated with IL-1µ, PBM with 810 nm alone or combined with 660 nm or 980 nm showed more potent inhibition of INF-γ, IL-17A/F, monocyte chemoattractant protein 1 (MCP-1), and IL-6 as compared to 660 nm + 980 nm irradiation, demonstrating the beneficial effect of PBM with 810 nm in regulating inflammatory responses [99]. The anti-inflammatory effect of PBM was also noted on 5-fluorouracil-induced oral mucositis in hamsters via inhibiting levels of NF-κβ and TNF-α. Interestingly, better outcomes with reduced inflammation and shorter repair time of oral mucositis were found in groups treated with lower energy (0.24 J) rather than high energy (1 J), again demonstrating the biphasic effect of PBM dose [100].

In addition, PBM attenuates inflammation via modifying the phenotype of immune cells. Macrophages, as one type of innate immune cell, are usually categorized into two types, M1-like and M2-like macrophages. M1 macrophages are primarily associated with pro-inflammation immune responses, while M2 cells are potently anti-inflammatory [101]. PBM (945 nm, 3 J/cm^2^) significantly reduced the gene expressions of IL-6 and TNF-α in isolated macrophages incubated with Bothrops jararacussu venom (BjV). Furthermore, this study indicated that PBM increased the polarization of macrophages to the M2 phenotype, repairing tissue and suppressing inflammation [102]. Liao et al. also reported that PBM inhibited the inflammatory response through the macrophage M2 polarization in human THP-1 monocyte cells, human primary macrophages, and mice intraperitoneal macrophages, which was regulated by mitochondrial respiratory gene citrate synthase [103]. In addition, in the LPS-induced rat model of acute lung injury, PBM (660 nm, 5 J/cm) inhibited the inflammatory response by reducing the neutrophils infiltration into the bronchoalveolar lavage and myeloperoxidase activity, which was associated with decreased secretion of inflammatory interleukins, such as IL-1β, IL-6, and IL-17 [104]. A recent study further confirmed the anti-inflammatory effect of PBM on macrophages collected from male mice with reduced pro-inflammatory cytokines and increased anti-inflammatory cytokines. Interestingly, the biphasic dose-response was observed in cell viability, which was increased at a lower dosage but decreased at a higher dosage [105].

### 3.4. PBM and Gut Microbiota Dysbiosis

Accumulating evidence suggests that PBM can restore dysregulated gut microbiome by improving the bacteria’s composition, abundance, and function [17,82]. Bicknell et al. investigated the potential effect of PBM in gut microbiota modulation. They found that PBM at 660 and 808 nm altered the microbiome diversity in healthy Balb/c mice with the increased abundance of the *Allobaculum,* a bacterium associated with a healthy mouse microbiome [17]. Moreover, PBM has beneficial effects on various neurodegenerative diseases via regulating gut microflora [82,106]. For example, in amyloid precursor protein/presenilin 1 (APP/PS1) transgenic mice (an Alzheimer’s disease (AD)-like mouse model), the application of PBM (peak wavelength 7.7–10 µm) for six weeks reversed the gut microbiota composition in the APP/PS1 mice, suggesting that the positive effect of PBM for AD treatment is likely via the brain-gut axis [107]. In line with this finding, Chen et al. reported that PBM at wavelengths of 630 nm, 730 nm, and 850 nm for eight weeks improved the AD condition in an amyloid-β (Aß)-induced AD mouse model. Finally, they found that PBM increased the diversity and abundance of *Helicobacter*, uncultured *Baterioidales,* and decreased *Rikenella* in AD mice. These findings further confirmed that gut flora-targeted PBM provided the therapeutic potential for AD treatment by restoring the dysbiosis of gut flora [82]. Recently, Bicknell and his team found that PBM can potentially change the microbiome community in patients with Parkinson’s disease (PD), leading to an increased abundance of *Bacteroides, Macellibacteriodes, Barnesiella, Odoribacter*, *Alistipes,* and some unidentified Bacteroidales after a 12-week PBM treatment regimen [85].

## 4. PBM Therapies in Kidney Diseases

Several preclinical and clinical studies with PBM application have suggested positive outcomes on CKD (Table 1). For example, the renoprotective effect of PBM has been demonstrated in a rat model of metabolic syndrome-induced CKD, which was associated with reduced blood pressure, increased glomerular filtration rate, and decreased tubulointerstitial fibrosis [108]. Oron et al., reported that the administration of PBM (804 nm; 1 J/cm^2^) to areas remote from the kidney, specifically the tibia of rats, significantly improved pathological changes and kidney function. They suggested that promoted stem cells in bone marrow to migrate to injured kidneys, thus providing an indirect mechanism to improve kidney function [109].

PBM has shown beneficial effects in CKD via stimulating mitochondrial activity, improving mitochondrial dynamics, and ATP production. For example, in a streptozotocin-induced type 1 diabetic rat model, PBM (670 nm, 10.5 J/cm^2^) up-regulated the levels and activity of CCO, CAT, sodium-potassium ATPase and inhibited the production of the DNA adduct 8-hydroxy-2′-deoxyguanosine in the diabetic kidneys, thus contributing to improved renal function with reduced serum levels of blood urine nitrogen (BUN) and creatinine [110]. Furthermore, O’Connor et al. reported that laser irradiation (405, 532, and 635 nm; 3.662 J/cm^2^) synergized with mesenchymal stem cells (MSCs) significantly enhanced mitochondrial activity and reduced apoptosis in a mouse model of UUO. The laser at 532 nm combined with MSCs induced proliferative activity in the kidney cortex, while increased endothelial proliferation was observed when exposed to the 635 nm laser alone or the 405 nm synergized with MSC [111].

PBM can restore cellular homeostasis via balancing ROS and antioxidants. In a chicken model of TCDD (2,3,7,8-Tetrachlorodibenzo-p-dioxin)-induced kidney damage, Lim et al. found that PBM (670 nm, 4 J/cm^2^) significantly reversed the level of ATP and glutathione (GSH) contents decreased by TCDD. Moreover, PBM mitigated TCDD-induced oxidative stress in the chicken kidney as evidenced by enhanced activities of the antioxidant enzymes, including SOD, glutathione peroxidase (GPx), glutathione-S-transferase (GST), and glutathione reductase (GRx), as well as the increased level of lipid peroxidation [112]. Similarly, a study on ischemia/reperfusion (IR)-induced renal damage in diabetic rats showed that PBM (685 nm, 3.2 J/cm^2^) attenuated kidney injuries caused by renal IR, which was associated with an enhanced antioxidant defense system, reduced lipid peroxidation, and neutrophil infiltration. [113]. A recent study further confirmed that the protective effect of PBM was mediated through the regulation of oxidative stress in γ radiation-induced kidney damage. PBM (870 nm, 1.2 J/cm^2^) significantly attenuated γ radiation-induced oxidative stress with lower levels of H_2_O_2_, 4- hydroxynonenal, and total oxidative status in kidneys. In addition, PBM markedly stimulated the activity of the detoxification enzyme GSH-T and increased expressions of GSH, SOD, GPx, GRx, and CAT, leading to the improvement in the corpuscles and tubules of the kidney [114].

PBM can also exert its overall anti-inflammatory capacity by reducing pro-inflammatory cytokines and inflammatory mediators. In a rat model of UUO, PBM (780 nm, 22.5 J/cm^2^) reduced inflammatory markers MCP-1, IL-6, and pro-fibrotic factors TGF-β1 and Smad3, indicating that PBM protected UUO-induced interstitial fibrosis by mitigating the inflammation. [115]. The anti-inflammatory effect of PBM was also confirmed in a rat model of crescentic glomerulonephritis. The results showed that PBM (830 nm, 220 J/cm^2^) significantly reduced inflammatory cytokines such as IL-1β and TNF-α in the kidney cortex. Histological results also found that PBM inhibited the crescent formation and accumulation of ED+ macrophage and CD8+ lymphocytes in the glomeruli [116].

Intravascular laser irradiation of blood, via an intravenous needle or transcutaneous venous irradiation [118], has been reported as having beneficial effects for the treatment of acute kidney injury [117]. The results confirmed that PBM (405 nm; Output power 1.5 mW for 30 min) had benefits in improving kidney function in patients with AKI, which was with a significant reduction in serum and urine neutrophil gelatinase-associated lipocalin. Moreover, PBM has also shown positive outcomes in CKD patients such as improvement in muscle (handgrip) strength [119] or salivary glands [120] of CKD patients on hemodialysis.

PBM may also play an important role in attenuating kidney disease via the gut-kidney axis, as the gut-kidney axis is increasingly considered a crucial contributor to CKD [2]. A disrupted gut microbiome worsens the gut leakage in patients with CKD, leading to translocation of gut bacteria and endotoxin through the gut barrier due to disruption of tight junctions and elevated uremic toxins, which accelerates the deterioration of CKD [121,122]. Consequently, kidney function loss causes the increased retention of uremic toxins secreted into the gut lumen, driving up the severity of gut microbiota dysbiosis. Therefore, it is logical to attempt interventions to break this vicious cycle between kidneys and gut. PBM, as discussed above, may favorably restore the gut microbiota, which might contribute to increased gut microbiota produced SCFAs and strengthen the gut barrier with decreased uremic toxins and endotoxins [17].

Directly targeting the kidney in adult humans requires penetration of the PBM through skin, muscle, some bone, and fat (perirenal capsule) to a depth of a number of cm. It would be expected that few photons will reach the kidney [76], although near infrared light is known to have deeper penetration than visible light [123,124,125]. Photons not reaching the kidney will be absorbed by muscle and fat, which are known to reduce inflammation [126]. A nephrostomy catheter or a catheter passed through the urethra, the bladder and ureter to the kidney might be used to increase the light penetration to the kidney. However, they are quite invasive and would be challenging for repeated PBM treatment. Interestingly, increasing evidence has suggested that, in addition to the direct effect of PBM, PBM can also provide protective effect through indirect systemic effects such as immune/inflammatory response or activation of secondary messenger pathways and transcription factors [127,128,129]. There are numerous examples of the benefits of indirect PBM application (that is PBM not specifically directed towards the site of the injury/disease) in both animal models and humans, including in cardiovascular disease [130,131], diabetic retinopathy [132], Alzheimer’s disease [133], Parkinson’s disease [134,135], in lung injury [104,136] and potentially depression [137]. Thus, there are arguments for combining direct (targeted) and indirect (remote) PBM for therapeutic benefit [138]. Clearly more studies are needed to understand the mechanistic basis of PBM to provide an opportunity to use this non-pharmacological therapy for CKD.

In summary, PBM might exert its beneficial effect by targeting kidney directly or indirectly (PBM being applied remote from the kidney area) through systemic effects. As shown in Figure 1, PBM might have the potential to improve kidney function via promoting mitochondrial activity, suppressing oxidative stress, inhibiting inflammation, and restoring gut microbiota homeostasis.

## 5. Conclusions and Future Studies

Despite the promising effects of PBM, outcomes of previous studies and clinical PBM treatments are varied. There is no universal agreement on wavelength, energy density, treatment duration, and administration mode, resulting in obstacles to the clinical application of PBM. In addition, anatomical interference is another barrier to the delivery of PBM to treat internal organs. Importantly, due to the biphasic feature of PBM, the energy density has to be tailored for different diseases, together with the establishment of specific standard operating procedures and practical treatment predictors.

Thus far, there are no evidence-based guidelines on the protocols for delivering PBM therapy in the treatment of CKD. Hence parameters of PBM, the location of PBM irradiation, and administration approaches vary. Although there are some concerns about whether enough light will reach kidney, it’s postulated that the beneficial effects of PBM might not entirely depend on direct influence of the light, but also some indirect systemic effects through the blood or cellular mechanisms. Hence, future standardized studies are warranted to determine the therapeutic effect of PBM in CKD and better understand its underlying mechanisms, which is essential for its successful clinical translation. Moreover, advanced techniques in detecting or testing target markers, such as oxidative species and target PBM energy density, are urgently needed to optimize PBM treatment.

In conclusion, the mechanisms of action of PBM include modulating mitochondrial dysfunction, oxidative stress, inflammation, and gut microbiota dysbiosis, all of which are involved in the onset and progression of CKD. Therefore, PBM has the potential to have a positive effect in mitigating CKD by targeting these mechanisms.

## Figures and Tables

**Figure 1 ijms-23-08043-f001:**
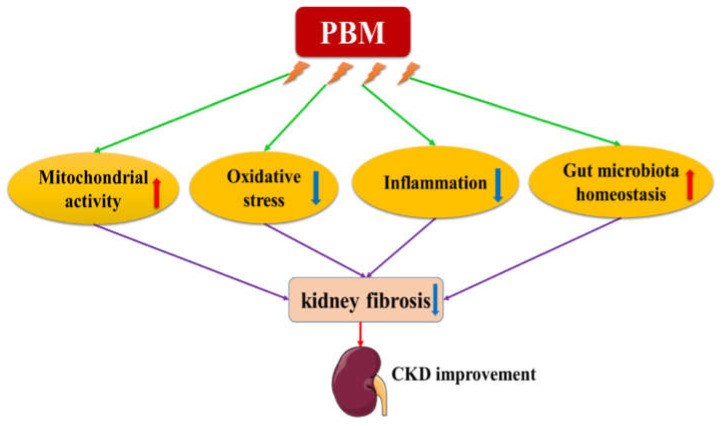
Potential mechanisms of the therapeutic effects of PBM on CKD. PBM may improve kidney function by promoting mitochondrial activity, suppressing oxidative stress, inhibiting inflammation, and restoring gut microbiota homeostasis.

**Table 1 ijms-23-08043-t001:** PBM studies in kidney diseases.

Model of Disease	PBM Parameters	Effects	Reference
MS-related KI (rat model)	Wavelength: 785 nm; Energy density:12 J/cm^2^; Exposure time: 20.4 s; Output power: 5 mW; Applied five times a week for eight weeks	PBM reduced blood pressure, increased glomerular filtration rate, and decreased tubulointerstitial fibrosis in rats with metabolic syndrome.	[108]
IRI (rat model)	Wavelength: 804 nm; Energy density:1 J/cm^2^; Exposure time 100 s; Output power: 10 mW/cm^2^; Laser was applied immediately post-IRI and again at 24 h post-IRI	The irradiation of PBM to bone morrow following IRI injury improved pathological changes and kidney function.	[109]
Diabetic model (rat model)	Wavelength: 670 nm; Energy density:10.5 J/cm^2^; Exposure time: 300 s; Output power: 35 mW; Applied 5 times a week for 14 weeks	PBM reduced serum levels of BUN and creatinine, which were associated with increased catalase activity and expression as well as sodium-potassium ATPase activity in all light-treated diabetic rats. The activity and expression of glutathione peroxidase and the expression of sodium-potassium ATPase were unchanged with PBM irradiation.	[110]
UUO (mouse model)	Wavelength: 635,532 and 405 nm; Energy density:3.6 mJ/cm^2^; Exposure time 300 s; Output power: 17.5 mW; Applied once a week for one week	PBM at three wavelengths synergized with mesenchymal stem cells (MSCs) to promote mitochondrial function and reduce apoptosis. Proliferative activities were enhanced in the kidney cortex after PBM irradiation at 532 nm, followed by MSC administration. PBM at 635 alone or synergized with MSCs significantly increased endothelial proliferation.	[111]
TCDD-treated kidney injury model (Chicken model)	Wavelength: 670 nm; Energy density: 4 J/cm^2^; Output power: 6 mW; Applied once per day for 20 days	PBM attenuated chemical-induced renal injury via inhibiting oxidative stress and restoring ATP production.	[112]
Diabetic Nephropathy (rat model)	Wavelength: 685 nm; Energy density:3.2 J/cm^2^; Exposure time:60 s; Output power: 15 mW; Intraoperative single dose at 15 min after UUO	The diabetic rats treated with PBM showed a significant improvement in glutathione, SOD, and CAT levels and a significant decrease in MPO level compared to the nephrectomy group.	[113]
Post irradiation model (rat model)	Wavelength: 870 nm; Energy density:1.2 J/cm^2^; Exposure time: 30 s/each 6 points; Output power: 100 mW; Applied once a week for one week	PBM attenuated the oxidative stress and improved the histological changes in the liver and kidney tissues of ionizing radiation-induced mental disorder.	[114]
UUO (rat model)	Wavelength: 780 nm; Energy density:22.5 J/cm^2^; Exposure time:30 s; Output power: 30 mW; Applied once a week for two weeks	PBM reduced UUO-induced interstitial fibrosis by reducing inflammation in the UUO rat models.	[115]
Crescentic glomerulonephritis (rat model)	Wavelength: 830 nm; Energy density: 20 J/cm^2^; Output power: 250 mW; Applied once a week for two weeks	PBM suppressed crescent formation and infiltration of ED1+ macrophages and CD8+ lymphocytes into the glomeruli as well as the levels of IL-1β and TNF-α messenger RNA in the kidney cortex.	[116]
AKI-patients	Wavelength: 405 nm; Output power: 1.5 mW for 30 min; Applied on days 1, 3 and 5.	PBM improving kidney functions in patients with AKI as evidenced by reduced the serum and urine neutrophil gelatinase-associated lipocalin (NGAL).	[117]

Abbreviations: MS: metabolic syndrome; KI: kidney injury; BUN: blood urea nitrogen; UUO: unilateral ureteric obstruction; IRI: ischemic reperfusion kidney injury; TCDD: 2,3,7,8-Tetrachlorodibenzo-p-dioxin; SOD: superoxide dismutase; CAT: catalase; MPO: myeloperoxidase; IL-1β: interleukin-1β; TNF-α: tumor necrosis factor-α; NGAL: neutrophil gelatinase-associated lipocalin.

## Data Availability

Not applicable.

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
