# Peer review of "Therapeutic Potential of Photobiomodulation for Chronic Kidney Disease"

_ijms, 2022, doi:10.3390/ijms23148043_

Round 1

Reviewer 1 Report

This is a timely and interesting contribution to the problem of chronic kidney disease. I have the following remarks:

1. In the introduction please add a few more details for the general "Int. J. Molecular Sciences" reader. For instance you may note that CKD is often only discovered when analysing a patient's blood or urine for another problem, i.e., the patient may not be aware of the disease. Also please describe possible early symptoms, as well as the five stages of the disease, etc.

2. On page 1 lines 42,43 replace red light by visible light if you want to include the wavelengths from 450-1100 nm. 450 nm is certainly not red.

3. Page 4: Under 3. PBM overview. Line 194, 10-500 mW should probably be 10-500 mW/cm2? Also with 500 mW/cm2 you may start heating the tissue. So please clarify your units. 

4. A great weakness of PBM (or low level light therapy) is that in the literature one can never find a proper definition of optimal conditions. This is terrible as the PBM effect seems to be bi-phasic. i.e. There does not seem to be a proper definition of the optimal wavelengths to be used, of the optimal light intensity, of the time of exposure, and the number of times irradiation is to take place at which intervals. Please make this very clear as it significantly weakens the case  for a scientific approach to PBM as different laboratories seem to use different conditions.

5. In the present case you wish to irradiate (presumably) a whole kidney, roughly the size of a human fist. How do you propose to do this in a way where the light dose is at least more our less homogeneous throughout the kidney, or does that not matter?

6. Another problem that absolutely needs to be discussed in this article, is the problem of how to get the light for the PBM to the kidney. The distance from skin to calyx may be typically of the order of 8 cm in a grown person, and that distance is too far for reasonable light penetration. A nephrostomy catheter may be used but that is already quite invasive and problematic for repetitive illumination over days/weeks. The same holds for a catheter passed through the urethra, the bladder and ureter to the kidney. Please clarify what approach you recommend.

7. On page 5 line 230, what is meant by reducing intra-mitochondrial water viscosity reduction?

8. Maybe a remark should be made that on some CKD patients PBM is already used, but the irradiation is for instance done on the salivary glands or the hands of these patients or the lower limbs, rather than direct irradiation of the kidneys. 

Author Response

Point 1: In the introduction please add a few more details for the general "Int. J. Molecular Sciences" reader. For instance you may note that CKD is often only discovered when analysing a patient's blood or urine for another problem, i.e., the patient may not be aware of the disease. Also please describe possible early symptoms, as well as the five stages of the disease, etc.

Response 1: More detail information about CKD has been included in the revised manuscript as suggested.

Point 2: On page 1 lines 42,43 replace red light by visible light if you want to include the wavelengths from 450-1100 nm. 450 nm is certainly not red.

Response 2: Red light in lines 42,43 were replaced by visible light as suggested.

Point 3: Page 4: Under 3. PBM overview. Line 194, 10-500 mW should probably be 10-500 mW/cm2? Also with 500 mW/cm2 you may start heating the tissue. So please clarify your units. 

Response 3: We have double checked and the units in the manuscript are consistent with that used in other literatures. The relevant references have been included in the revised manuscript.

Point 4: A great weakness of PBM (or low level light therapy) is that in the literature one can never find a proper definition of optimal conditions. This is terrible as the PBM effect seems to be bi-phasic. i.e. There does not seem to be a proper definition of the optimal wavelengths to be used, of the optimal light intensity, of the time of exposure, and the number of times irradiation is to take place at which intervals. Please make this very clear as it significantly weakens the case for a scientific approach to PBM as different laboratories seem to use different conditions.

Response 4:  We have included more information about the biphasic dose response of PBM to make it clear that no optimal conditions for effective PBM doses have been agreed upon. Appropriate irradiation doses in a practical PBM application should be determined according to the specific condition.

Point 5: In the present case you wish to irradiate (presumably) a whole kidney, roughly the size of a human fist. How do you propose to do this in a way where the light dose is at least more our less homogeneous throughout the kidney, or does that not matter?

Response 5: PBM works on the principle that the light energy (photons) penetrates tissue where it triggers photochemical changes. The well accepted underlying mechanism of PBM is that the photon energy is primarily absorbed by light-sensitive molecules known as chromophores, mainly present inside mitochondria. Due to this mechanism of action of PBM, it seems that it impossible to deliver a homogeneous light throughout the kidney. However, it will not affect the beneficial effect of PBM as suggested by the studies in kidney disease (Table 1). More work is needed to better understand its underlying mechanisms, which is essential for its successful clinical translation.

Point 6: Another problem that absolutely needs to be discussed in this article, is the problem of how to get the light for the PBM to the kidney. The distance from skin to calyx may be typically of the order of 8 cm in a grown person, and that distance is too far for reasonable light penetration. A nephrostomy catheter may be used but that is already quite invasive and problematic for repetitive illumination over days/weeks. The same holds for a catheter passed through the urethra, the bladder and ureter to the kidney. Please clarify what approach you recommend.

Response 6: As discussed in the revised manuscript, near-infrared lasers have lower scattering factors and much higher penetration through the skin and soft/hard tissue and therefore are considered to be suitable for deep-sitting organs. In addition, the skin contact technique will also increase the penetration into tissues as it reduces reflection to provide greater power density in the target tissue. It is also worth mentioning that the beneficial effects of PBM might not entirely depend on direct influence of the light, but also some indirect and systemic effects through the blood or cellular mechanisms. More research to apply PBM to the skin surface of the kidney is needed which will provide a non-invasive approach for the treatment of CKD. The relevant information has been included in the revised manuscript.

Point 7: On page 5 line 230, what is meant by reducing intra-mitochondrial water viscosity reduction?

Response 7: Sorry for the confusion. It has been changed to “by reducing the viscosity of interfacial water layers (IWL)” to make it clearer. IWL is the thin layer of water molecules on the inner side of the internal mitochondrial membrane. The information has been amended in the revised manuscript.

Point 8: Maybe a remark should be made that on some CKD patients PBM is already used, but the irradiation is for instance done on the salivary glands or the hands of these patients or the lower limbs, rather than direct irradiation of the kidneys. 

Response 8: The suggested information has been included in the Page 10 of the revised manuscript.

Reviewer 2 Report

1. Future directions/implications should be separated prior to conclusion

2. Below the Table 1, should add all abbreviations that are used in table.

3. Figure 1 needs high/better resolution quality of figure.

4. Writing should be more formal. For example, etc. should not be used in the conclusion. In addition, reference to Table 1 should not be included in the conclusion. 

Author Response

Point 1: Future directions/implications should be separated prior to conclusion

Response 1: Future directions/implications were separated prior to conclusion as suggested.

Point 2: Below the Table 1, should add all abbreviations that are used in table.

Response 2: All abbreviations were added below table 1 as suggested.

Point 3:  Figure 1 needs high/better resolution quality of figure.

Response 3: Figure 1 with high resolution were added and will submitted in the submission system.

Point 4:  Writing should be more formal. For example, etc. should not be used in the conclusion. In addition, reference to Table 1 should not be included in the conclusion. 

Response 4: Writing issues have been fixed as suggested.

Reviewer 3 Report

There is not a single reference to the studies of Russian authors, although in Russia low-level laser therapy for kidney diseases has been successfully used for almost 50 years.

There are no recommendations on the methods of low-level laser therapy, what parameters are needed for illumination on the projection of the kidneys and other parts, there is no mention of intravenous laser blood illumination in the modern version (wavelengths of 635, 525 and 365 nm) - without this technique, in principle, treatment is futile.

Razzaghi M.R., Ghanei E., Sheida Malekian S., Mazloomfard M.M. Intravenous laser therapy in patients with acute kidney injury: A randomized clinical trial // J Lasers Med Sci. – 2021; 12:e49. doi: 10.34172/jlms.2021.49.

The review also did not cite these interesting publications.

Pavesi V.C.S., Martins M.D., Coracin F.L. et al. Effects of photobiomodulation in salivary glands of chronic kidney disease patients on hemodialysis // Lasers in Medical Science. – 2021; 36(6): 1209–1217. doi: 10.1007/s10103-020-03158-0.

Macagnan F.E., Baroni B.M., Cristofoli E.Z. et al. Acute effect of photobiomodulation therapy on handgrip strength of chronic kidney disease patients during hemodialysis // Lasers in Medical Science. – 2019; 34(4): 835-840. doi: 10.1007/s10103-018-2593-7.

Oron U., Tuby H., Maltz L. et al. Autologous bone-marrow stem cells stimulation reverses post-ischemic-reperfusion kidney injury in rats // Am J Nephrol. – 2014; 40(5): 425-433. doi: 10.1159/000368721

Author Response

Point 1: There is not a single reference to the studies of Russian authors, although in Russia low-level laser therapy for kidney diseases has been successfully used for almost 50 years.

Response 1: Thanks for your suggestion. More studies from Russian authors have been included in the revised manuscript (Ref,12, Ref 99, Ref 121).

Point 2: There are no recommendations on the methods of low-level laser therapy, what parameters are needed for illumination on the projection of the kidneys and other parts, there is no mention of intravenous laser blood illumination in the modern version (wavelengths of 635, 525 and 365 nm) - without this technique, in principle, treatment is futile.

Response 2:  So far, there are no agreed recommendations on the methods of PBM therapy in CKD. This review aims to summarize the potential mechanisms and studies focusing on PBM and CKD, which hope to inspire more studies in this field. The study using intravascular laser irradiation of blood for the treatment of kidney disease is interesting and now included in the revised manuscript (REF 120).

Point 3: references suggested below

  1. Razzaghi M.R., Ghanei E., Sheida Malekian S., Mazloomfard M.M. Intravenous laser therapy in patients with acute kidney injury: A randomized clinical trial // J Lasers Med Sci. – 2021; 12:e49. doi: 10.34172/jlms.2021.49.
  2. Pavesi V.C.S., Martins M.D., Coracin F.L. et al. Effects of photobiomodulation in salivary glands of chronic kidney disease patients on hemodialysis // Lasers in Medical Science. – 2021; 36(6): 1209–1217. doi: 10.1007/s10103-020-03158-0.
  3. Macagnan F.E., Baroni B.M., Cristofoli E.Z. et al. Acute effect of photobiomodulation therapy on handgrip strength of chronic kidney disease patients during hemodialysis // Lasers in Medical Science. – 2019; 34(4): 835-840. doi: 10.1007/s10103-018-2593-7.
  4. Oron U., Tuby H., Maltz L. et al. Autologous bone-marrow stem cells stimulation reverses post-ischemic-reperfusion kidney injury in rats // Am J Nephrol. – 2014; 40(5): 425-433. doi: 10.1159/000368721

Response 3:  Thank you for the suggestion. All the recommended references have been included in the revised manuscript.

Round 2

Reviewer 1 Report

The manuscript is now improved but still needs some further precisions, especially concerning the light dosimetry:

1. the 10-500 mW indication in the paper is not usable scientifically. You absolutely have to specify to what surface you apply this light. Light in nearly all photo medicine is applied in Watts (or mW) per cm2. Over time when applied per second that becomes Joules (or mJ) per cm2. 10 or 500 mW could be the output of the light source, but you apply it to a certain surface. Furthermore as I mentioned previously, certain wavelengths applied at 500 mW per cm2 can start heating the tissue. If you just apply the light to the skin, as you seem to be implying, and do this via an optical fiber, than with 500 mW coming out of a fiber in contact with the skin you would probably burn the skin.

2. As I mentioned previously, the problem for PBM of the kidney is how to do the irradiation, as the kidney is many cm below the skin surface, and on top of that the kidney itself is a many cm sized organ. You absolutely should consult with a medical physicist on how you want to do this irradiation (I have already previously have given you some indications what the overall possibilities are). Typically, if you would irradiate at the skin surface using the 450 nm light you mention in your paper, there would be essentially no light of that wavelength reaching the kidney. Even the near IR wavelength penetrate several cm  of tissue with great losses! So you will have to deliver the light in one of the ways I mentioned in my previous comments, and you will have to discuss this properly in your paper. Most people in medical physics that are familiar with laser applications can help you with this.

3. I am not sure at all what is meant by the alternative mechanism you talk about in which the viscosity of water is caged in a very thin layer near the inner side of the inner membrane of the mitochondria. Please describe this mechanism in enough detail that the average reader of this journal can understand it.

Author Response

  1. the 10-500 mW indication in the paper is not usable scientifically. You absolutely have to specify to what surface you apply this light. Light in nearly all photo medicine is applied in Watts (or mW) per cm2. Over time when applied per second that becomes Joules (or mJ) per cm2. 10 or 500 mW could be the output of the light source, but you apply it to a certain surface. Furthermore as I mentioned previously, certain wavelengths applied at 500 mW per cm2 can start heating the tissue. If you just apply the light to the skin, as you seem to be implying, and do this via an optical fiber, than with 500 mW coming out of a fiber in contact with the skin you would probably burn the skin.

      Response 1: The sentence has been modified to avoid the confusion (Line 195,196).

  1. As I mentioned previously, the problem for PBM of the kidney is how to do the irradiation, as the kidney is many cm below the skin surface, and on top of that the kidney itself is a many cm sized organ. You absolutely should consult with a medical physicist on how you want to do this irradiation (I have already previously have given you some indications what the overall possibilities are). Typically, if you would irradiate at the skin surface using the 450 nm light you mention in your paper, there would be essentially no light of that wavelength reaching the kidney. Even the near IR wavelength penetrate several cm of tissue with great losses! So, you will have to deliver the light in one of the ways I mentioned in my previous comments, and you will have to discuss this properly in your paper. Most people in medical physics that are familiar with laser applications can help you with this.

Response 2: A paragraph (Line 394-413) that discusses how PBM exerts its effect on the kidney has been included in the revised manuscript as suggested, including direct and indirect PBM, infrared wavelengths and using catheters.

  1. I am not sure at all what is meant by the alternative mechanism you talk about in which the viscosity of water is caged in a very thin layer near the inner side of the inner membrane of the mitochondria. Please describe this mechanism in enough detail that the average reader of this journal can understand it.

Response 3: As the alternative mechanism has not been well explained, the information has been removed to avoid any confusion.

Reviewer 3 Report

The manuscript has been significantly improved

Author Response

Thank you so much for your comments, which enrich my knowledge in PBM field.